# Exosomal microRNA Differential Expression in Plasma of Young Adults with Chronic Mild Traumatic Brain Injury and Healthy Control

**DOI:** 10.3390/biomedicines10010036

**Published:** 2021-12-24

**Authors:** Rany Vorn, Maiko Suarez, Jacob C. White, Carina A. Martin, Hyung-Suk Kim, Chen Lai, Si-Jung Yun, Jessica M. Gill, Hyunhwa Lee

**Affiliations:** 1National Institute of Nursing Research, National Institutes of Health, Bethesda, MD 20814, USA; rany.vorn@nih.gov (R.V.); carina.martin@nih.gov (C.A.M.); kimhy@mail.nih.gov (H.-S.K.); laichi@mail.nih.gov (C.L.); 2School of Medicine, University of Nevada, Las Vegas, NV 89102, USA; suarem2@unlv.nevada.edu; 3College of Liberal Arts, University of Nevada, Las Vegas, NV 89154, USA; whitej31@unlv.nevada.edu; 4Yotta Biomed, LLC, Bethesda, MD 20817, USA; sijungyun@yottabiomed.com; 5School of Nursing and Medicine, Johns Hopkins University, Baltimore, MD 21205, USA; JessicaGill@jhu.edu; 6Center for Neuroscience and Regenerative Medicine, Uniformed Services University of the Health Science, Bethesda, MD 20814, USA; 7School of Nursing, University of Nevada, Las Vegas, NV 89154, USA

**Keywords:** mild traumatic brain injury, microRNA, exomiRNA, exosome

## Abstract

Chronic mild traumatic brain injury (mTBI) has long-term consequences, such as neurological disability, but its pathophysiological mechanism is unknown. Exosomal microRNAs (exomiRNAs) may be important mediators of molecular and cellular changes involved in persistent symptoms after mTBI. We profiled exosomal microRNAs (exomiRNAs) in plasma from young adults with or without a chronic mTBI to decipher the underlying mechanisms of its long-lasting symptoms after mTBI. We identified 25 significantly dysregulated exomiRNAs in the chronic mTBI group (*n* = 29, with 4.48 mean years since the last injury) compared to controls (*n* = 11). These miRNAs are associated with pathways of neurological disease, organismal injury and abnormalities, and psychological disease. Dysregulation of these plasma exomiRNAs in chronic mTBI may indicate that neuronal inflammation can last long after the injury and result in enduring and persistent post-injury symptoms. These findings are useful for diagnosing and treating chronic mTBIs.

## 1. Introduction

More than 5.5 million mild traumatic brain injuries (mTBI) are reported annually in the United States [1]. In most individuals with mTBI, the symptoms can resolve within days to weeks; yet for 10–15% of mTBI patients, symptoms last longer than three months and can result in disability [2,3]. Repetitive mTBIs, which usually contribute to chronic mTBI with unresolved and persistent symptoms, are common in athletes and military personnel, and these individuals have a higher risk of chronic neurologic impairment [4,5]. Over 25% of individuals with long-term mTBI consequences are not able to return to work 1-year post-injury [1]. Chronic mTBI becomes a major health concern due to life-long disabilities and long-term consequences that severely compromise the affected individuals’ quality of life [6,7,8]. mTBI costs each patient $36,000 for rehabilitation [9] and the entire nation nearly $17 billion each year [10]. However, molecular mechanisms critical to chronic mTBI symptoms are currently unknown.

MicroRNAs (miRNAs) are small, single-stranded non-coding RNAs that regulate gene expression at the post-transcriptional level of target messenger RNA [11,12]. Circulating miRNAs are found in biofluids such as serum, plasma, saliva, and cerebrospinal fluid. They are stable and resistant to RNase digestion because they are encapsulated into extracellular vesicles called exosomes [13]. Exosomal miRNAs (exomiRNAs) in biofluids are potential biomarker candidates for diagnosis and prognosis of neurodegenerative disorders [14] such as Alzheimer’s disease (AD), Parkinson’s disease, and TBI due to their stability, and have the ability to regulate hundreds of target genes [13,15]. Dysregulated miRNAs may also reflect changes at the molecular level after the TBI [16].

We performed this comparative research to investigate differential expression of plasma exomiRNAs in the participants exhibiting chronic mTBIs such as repetitive head injuries compared with healthy controls. Identified plasma exomiRNAs were analyzed using bioinformatic knowledge-based Ingenuity Pathways Analysis (IPA) to understand the molecular pathway associated with plasma exomiRNAs after injury.

## 2. Materials and Methods

### 2.1. Study Protocol

Participants were recruited from the protocols granted by the Biomedical Institutional Review Board Committee at the University of Nevada, Las Vegas (UNLV) (Protocols No. 1048342 & 975928). Voluntary participants of this study were recruited via flyers and email advertisements addressing members of the UNLV campus, including the Military and Veteran Services Center and the Las Vegas community. They were voluntary participants aged 18 years or older with or without a self-reported history of mTBI. Those with mTBI histories were exposed to closed head trauma with loss of consciousness for less than 30 min and post-traumatic amnesia for less than 24 h, following the mTBI diagnosis guidelines defined by the American Congress of Rehabilitation Medicine [17]. The Neurobehavioral Symptom Inventory (NSI) and the Rivermead Post-Concussion Symptoms Questionnaire (RPQ) were used to assess the participant’s symptom experiences following mTBI. The NSI consists of 22 symptoms on a Likert scale of 0 to 4 (0 = none, 1 = mild, 2 = moderate, 3 = severe, and 4 = very severe), with a total sum score range of 0 to 88, and can be divided into three subscales affective, cognitive, and somatic/somatosensory [18]. The NSI is known to be reliable and valid for measuring post-concussive symptoms in TBI patients [19]. The RPQ is a 16-item questionnaire with a Likert scale of 0 to 4 (0 = none, 1 = mild, 2 = moderate, 3 = severe, and 4 = very severe) with a total sum score range of 0 to 64. The RPQ exhibits excellent internal consistency in TBI patients at all levels of severity—mild, moderate, and severe TBI [20]. Higher total scores of NSI and RPQ indicate more severe symptoms. Subjects with a previous or current diagnosis of neuropsychiatric disease (e.g., multiple sclerosis, attention deficit hyperactivity disorder, mood disorders, substance use disorders, etc.) or who were currently on any prescription drugs were excluded.

### 2.2. Plasma Collection

A phlebotomist drew blood in 4 mL blood collection tubes via venipuncture from each participant in the designated lab. Each 4 mL participant blood sample was centrifuged at 1900× *g* at 4 °C for 10 min. Once centrifuged, the plasma was carefully aspirated into DNase- and RNase-free Eppendorf tubes without disturbing the underlying buffy coat layer where most of the white blood cells were collected. The plasma samples were centrifuged in the Eppendorf tubes at 3000× *g* at 4 °C for an additional 15 min to remove cellular debris. The cleared supernatant was transferred to polypropylene cryovial tubes and stored at −70 °C in the UNLV Applied Biomedical Research Lab until purification.

### 2.3. ExomiRNA Purification

Plasma samples were removed from −70 °C storage, thawed, and centrifuged at 3000× *g* for 5 min to remove cryoprecipitates. The supernatant was transferred to a new tube to begin total RNA purification. Plasma exosomal total RNA purification was performed using the exoRNeasy Serum Plasma kit (Cat. # 77064, Qiagen, Hilden, Germany) according to the manufacturer’s protocol. In short, up to 1 mL or 4 mL of plasma was used, depending on the use of the Midi or Maxi kit, respectively. Both kits yielded similar output volumes. Plasma samples were washed with various buffer solutions provided by the manufacturer. Once washed, 700 μL of QIAzol lysis solution was added and centrifuged at 5000× *g* for 5 min to create a lysate solution. Then chloroform was combined with the lysate and centrifuged at 12,000× *g* for 15 min, after which the aqueous phase containing RNA was isolated. Total RNA purification was thereafter carried out using the RNeasy MinElute spin column, additional buffer solutions, and RNase-free water. Purified RNA samples were shipped overnight on dry ice to the Intramural Research Program laboratory at the National Institute of Nursing Research (NINR), in the National Institutes of Health (NIH) for additional processing.

### 2.4. ExomiRNA Profiling

Plasma exomiRNA analysis was performed in a nCounter MAX/FLEX Analysis System using nCounter^®^ Human v3 miRNA Expression Panels (NanoString Technologies, Seattle, WA, USA) that contained 798 unique miRNA probes. Probes for housekeeping genes such as ribosomal protein L10, beta-actin, beta-2-microglobulin, glyceraldehyde 3-phosphate dehydrogenase, and ribosomal protein L19, as well as endogenous miRNAs that were incorporated in the code sets, were used for analysis in addition to positive and negative controls. The complete RNA samples were prepared and run according to the manufacturer’s protocol. Counts of the reporter probes were obtained using the nCounter Digital Analyzer. Raw data were analyzed by nSolver Software version 4.0 (NanoString Technologies), and code count normalization was carried out by calculating the geometric mean of the ligation factors. Multiple testing correction with Benjamini-Hochberg’s False Discovery Rate (FDR; *p* < 0.100) was used as a cutoff for the differential dysregulation with statistical significance.

### 2.5. Network Analysis

The biological pathways associated with dysregulated plasma exomiRNAs were determined using Ingenuity Pathways Analysis (IPA) software (Ingenuity Systems Inc., Redwood City, CA, USA). All differentially expressed plasma exomiRNAs with FDR correction of less than 0.100 were uploaded into IPA for core pathway analysis. MicroRNA Target Filters were then used to identify the plasma exomiRNA-regulated mRNAs and enriched pathways. The Ingenuity Expert Findings, Ingenuity ExpertAssist Findings, miRecords, TarBase, and TargetScan Human source were used to assess gene target predictions, and miRWalk web tools were used for confirmation.

### 2.6. Statistical Analysis

Statistical analysis was conducted using SPSS version 28.0.0.0 (IBM Corp., Armonk, NY, USA). Demographic and clinical characteristics were compared between groups using Chi-square (*χ*^2^) and an independent sample *t*-test. The significance level was set at 0.05 in all tests.

## 3. Results

### 3.1. Demographics of the Study Population

A total of 40 participants, including 29 individuals with a history of mTBI and 11 without, participated in the study. The study participants were aged 19 to 36 years (24.8 ± 5.219). More than half of the study participants were female (52.5%) or White (57.5%). There were no statistical differences based on demographic characteristics, including age, gender, race, and BMI, between the two groups. In the chronic mTBI group (*n* = 29), 65.5% reported having more than 1 injury (41.4% with 2 or 3 injuries and 27.6% with 4 or more injuries), and the average number was 2.55 injuries (SD = 1.325). The average number of years since the last injury was 4.48 years (SD = 5.000). Only 1 participant in the mTBI group reported that they experienced both a brief loss of consciousness and post-traumatic amnesia resulting from the injury (3.4%). Two additional participants with a history of mTBI reported having post-traumatic amnesia (6.9%). The most common causes of injuries were sports-related activities (48.3%) (e.g., boxing, mixed martial arts [MMA] training, skating, football, etc.), followed by head hitting hard objects (e.g., sharp edge, metal materials, etc.) (20.7%), and high-level falls (17.2%). A few cases were related to military service activities (10.3%) or car accidents (3.4%). The demographics and clinical characteristics of the participants are presented in Table 1.

### 3.2. Differential Expression of Plasma ExomiRNAs

We assessed the expression levels of plasma exomiRNAs in the chronic mTBI group compared to control subjects. After normalization with ligation factors, we identified 25 plasma exomiRNAs differentially expressed in chronic mTBI compared to healthy control with an adjusted *p*-value < 0.100. Among them, 4 plasma exomiRNAs were upregulated, and 21 plasma exomiRNAs were downregulated in the chronic mTBI group compared with the control (Table 2).

### 3.3. Pathway Analysis

Pathway analysis using IPA revealed that dysregulated plasma exomiRNAs are related to neurological disease, organismal injury and abnormalities, and psychological disorders (Table 3). Top network analysis showed that these plasma exomiRNAs are associated with connective tissue disorders, inflammatory disease, organismal injury, and network abnormalities. The specific plasma exomiRNAs associated in this network (Figure 1) were hsa-miR-103-3p (synonym to hsa-miR-107), hsa-miR-126a-5p, hsa-miR-140-5p, hsa-miR-142-3p, hsa-miR-199a-3p, hsa-miR-221-3p (synonym to hsa-miR-222-3p), hsa-miR-23a-3p, hsa-miR-291a-3p (synonym to hsa-miR-520e), hsa-miR-374b-5p, hsa-miR-423-5p, hsa-miR-625-5p, hsa-miR-664-3p. These plasma exomiRNAs are directly or indirectly associated with acyl-CoA synthetase long-chain family member 6 (*ACSL6*), EPH receptor B6 (*EPHB6*), growth arrest specific 5 (*GAS5*), GNAS antisense RNA 1 (*Gnasas1*), hepatocellular carcinoma upregulated EZH2-associated long non-coding RNA (*HEIH*), homeobox A11 (*HOXA11*), phosphatase and tensin homolog (*PTEN*), resolvin D1, ribosomal protein S 15 (*RPS15*), tumor necrosis factor (*TNF*), vasohibin 1 (*VASH1*), and vascular endothelial growth factor A (*VEGFA*). Dysregulated exomiRNAs targeting mRNA associated with neuroinflammation pathways are presented in Figure 2 and Appendix A.

## 4. Discussion

In this study, we reported on several dysregulated plasma exomiRNAs in chronic mTBIs compared to healthy controls. We used multiplexed nCounter miRNA assays developed by NanoString technology to quantify low expression levels of exomiRNAs in plasma. The nCounter system is more sensitive than microarray for quantification of gene expression, with more accurate and reliable detection of very low expression levels [21]. The current study identified 25 dysregulated exomiRNAs that were associated with chronic mTBI. Pathway analysis showed that 14 plasma exomiRNAs were related to neurological disease, 23 plasma exomiRNAs were related to organismal injury and abnormalities, and 13 plasma exomiRNAs were related to psychological disease. Our results offered insights into the molecular mechanisms underlying injuries to brain function after mTBI.

In recent years, the role of miRNAs is indicated in the early diagnosis and progression of diseases including cancer, AD, and Parkinson’s disease [13,14,15,16]. Because they are short, miRNAs move freely across the blood-brain barrier (BBB) into the peripheral circulation, which can reflect changes in brain function due to a TBI [11,12,22]. Disruption of the BBB after injury promotes systemic inflammatory factors, neurotoxins, and pathogens into the brain and leads to neuronal damage [23]. Pathway analysis revealed that these plasma exomiRNAs are closely associated with *VEGFA*. VEGF is predominantly expressed in endothelial cells and plays an essential role in vascular development and neuroprotection [24,25]. Increased peripheral circulating levels of VEGF have been linked to alterations in BBB permeability following the infiltration of immune cells after a TBI [23,26]. Elevated concentrations of plasma VEGF and TNF-*α* protein were reported previously in mTBI patients [27].

Microglia and astrocytes are key mediators of neuroinflammation in the central nervous system (CNS) because of the release of inflammatory cytokines after brain injury. Ongoing neuroinflammation was observed when microglia became activated, even after 1 to 18 years of a single moderate-to-severe TBI [28,29,30]. Inflammatory cytokines were elevated in the plasma of mTBI patients within 24 h of injury [31], even up to a year later [32]. Many of the plasma exomiRNAs identified in our study were consistently found in the exosome and associated with neuroinflammation [5]. Previous studies reported that downregulation of hsa-miR-223-3p was associated with sporadic amyotrophic lateral sclerosis patients [33,34]. Downregulation of mitochondria-associated miR-142-3p was reported in a preclinical model of severe TBI [35]. As confirmed by target miRNAs and pathway analyses, these plasma exomiRNAs were found in brain cells and highly expressed in microglia [35], which are involved in neuroinflammation signaling and the glutamate signaling pathway through targeting the *SLC1A3* gene [36,37].

Our targeted analysis showed that downregulated hsa-miR-223-3p, has-miR-29b-3p, and has-miR-107, present in the chronic mTBI group, targeted the *NFIA* gene. *NFIA* is a member of the transcription factor nuclear factor I (NFI) family (*NFIB, NFIC*, and *NFIX*) and plays an essential role in neural development in the CNS [38]. NFIA is expressed in mature astrocytes and plays an essential role in retrieving forms of memory. A recent preclinical study showed that loss of *NFIA* inhibited neurotransmission and memory loss [39]. The major neurotransmitter in the CNS includes gamma-aminobutyric acid (GABA) and glutamate, which are involved in synapse plasticity, learning, and memory formation [40]. The targeted analysis showed that dysregulated has-miR-107 targeted the GABA receptor subunit gamma1 (*GABRG1*) and GABA A receptor subunit beta1 (*GABRB1*) genes. Mutation in the GABA_A_ receptor was associated with a neurological disorder and dysregulated GABA_A_ subunit gene expression as observed in the preclinical TBI model [41,42].

Downregulation of hsa-miR-107 (synonym to hsa-miR-103-3p) in our current study is consistent with previous findings [5]. miR-107 was reported as a synaptic region marker involved in the synaptogenesis signaling pathway targeting the synuclein gamma (*SNCG*), synaptotagmin 2 (*SYT2*), *SYT6*, and brain-derived neurotrophic factor (*BDNF*) genes (Appendix A) [43]. *SNCG* genes are predominantly expressed in neuronal tissue and play an important role in synaptic plasticity and dopamine regulation [44]. Overexpressed SNCG was associated with neurodegenerative disease pathogenesis in clinical and preclinical studies [45,46,47]. In addition to neurodegeneration, overexpressed SNCG was linked to poor prognosis in cancer and a preclinical TBI model [48,49]. A previous study showed that miR-107 was downregulated in AD and enhanced disease progression by regulating the beta-site amyloid precursor cleaving enzyme1 (*BACE1*) gene [50]. BACE1 is predominantly expressed in neurons and is responsible for the generation of amyloid-beta [51]. Overall, our findings suggest that chronic mTBI is associated with exomiRNA dysregulation, which may impact neurodegenerative disorders.

A major strength of this study is that the cases and controls were well matched. Although this study provides biological molecular insights into chronic mTBI, it was constrained by a small sample size. The majority of our sample population consisted of White individuals, which limits the genetic diversity and generalizability to all chronic mTBI populations. We were also unable to differentiate plasma exomiRNAs expression in single or multiple injuries due to our small sample size. Additional studies in a larger cohort should investigate the effect of single and repetitive injury on exomiRNAs expression level. Repetitive injury is a major risk for developing chronic neurological symptoms or impaired behaviors [52]. Future studies with larger cohorts will validate our observations that plasma exomiRNAs are associated with symptom deficits following brain injury.

In summary, chronic mTBI is associated with dysregulated exomiRNAs in plasma. These exomiRNAs were associated with inflammation processes that potentially link to neurological disability. Our data may aid in understanding the pathophysiological mechanism underlying the long-term impacts of mTBI. 

## Figures and Tables

**Figure 1 biomedicines-10-00036-f001:**
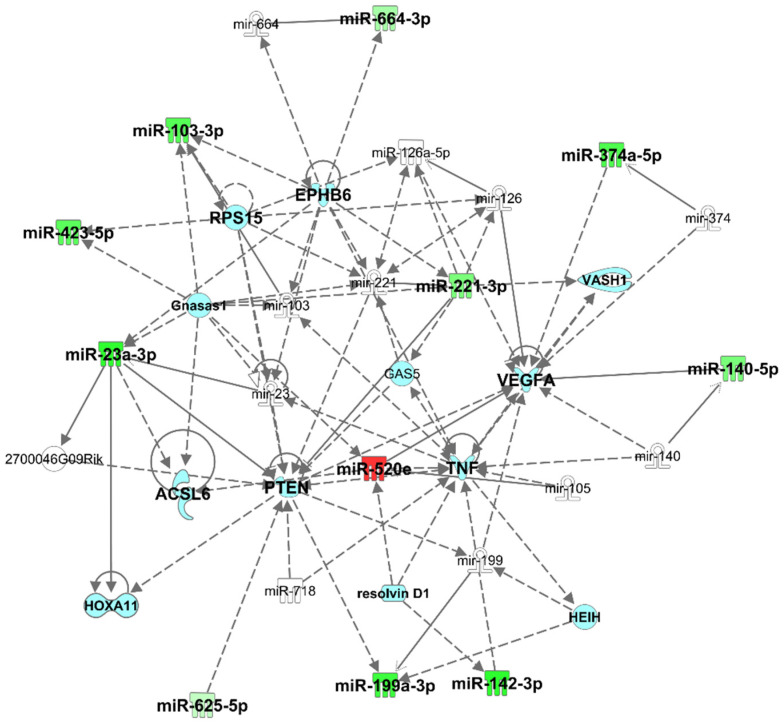
Top network identified by Ingenuity Pathway Analysis (IPA) for chronic mild traumatic brain injury (mTBI) versus control: Connective Tissue Disorders, Inflammatory Disease, and Organismal Injury and Abnormalities. Green indicates genes that are downregulated, and red indicates genes that are upregulated, where increased color saturation represents more extreme down- or upregulated in the dataset. Solid lines represent direct interactions, nontargeting interactions, or correlations between chemicals, proteins, or RNA. Dotted lines represented indirect interaction. Arrowed lines represent activation, causation, expression, localization, membership, modification, molecular cleavage, phosphorylation, protein–DNA interactions, protein–TNA interaction, binding regulation, and transcription. Shapes represent molecule type (double circle = complex/group; square = cytokine; diamond = enzyme; inverted triangle = kinase; triangle = phosphatase; oval = transcription regulator; trapezoid = transporter; circle = other). Reprinted from Ingenuity Pathway Analysis under a CC BY 4.0 license, with permission from QIAGEN Silicon Valley, original copyright 2000–2021.

**Figure 2 biomedicines-10-00036-f002:**
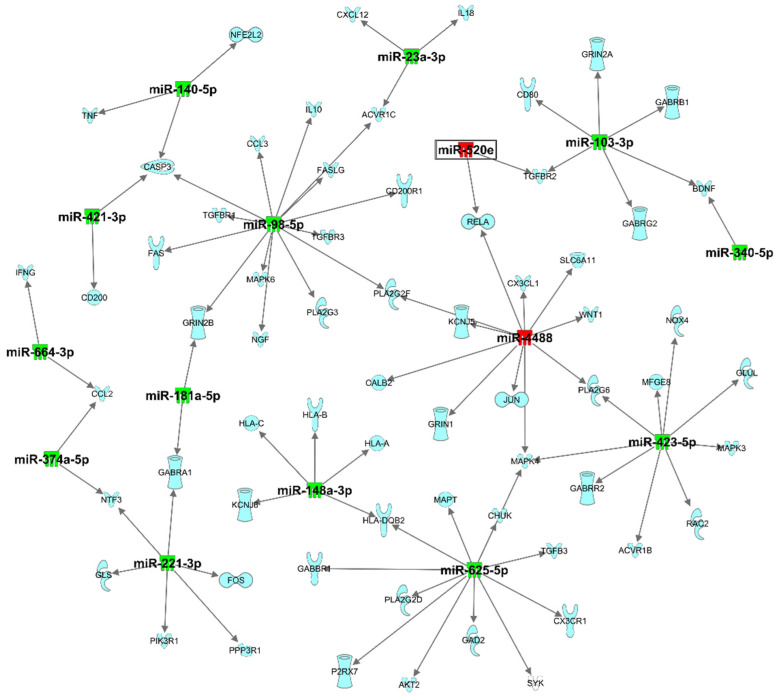
Target filter analysis of dysregulated exomiRNA targeting mRNA associated with neuroinflammation pathways. Green indicates genes that are downregulated, and red indicates genes that are upregulated, where increased color saturation represents more extreme down- or upregulated in the dataset. Reprinted from Ingenuity Pathway Analysis under a CC BY 4.0 license, with permission from QIAGEN Silicon Valley, original copyright 2000–2021.

**Table 1 biomedicines-10-00036-t001:** Demographic and Clinical Characteristics of Chronic mTBI and Healthy Control Participants (*n* = 40).

Characteristic	Overall (*n* = 40)	Chronic mTBI (*n* = 29)	Control (*n* = 11)	*χ*^2^ or *t*	*p*
Demographic					
Age, Mean (SD)	24.80 (5.22)	25.59 (5.36)	22.73 (4.41)	1.576	0.123
Gender, *n* (%)					
Males	19 (47.5)	15 (51.7)	4 (36.4)	0.755	0.488
Females	21 (52.5)	14 (48.3)	7 (63.6)		
Weight (kg), Mean (SD)	69.83 (14.64)	70.18 (13.29)	68.91 (18.45)	0.243	0.810
Height (cm), Mean (SD)	167.52 (10.94)	168.26 (10.48)	165.56 (12.38)	0.691	0.494
BMI, Mean (SD)	24.74 (3.77)	24.65 (3.40)	24.98 (4.81)	−0.247	0.807
Ethnicity/Race, *n* (%)					
Hispanic	6 (15.0)	4 (13.8)	2 (18.2)	1.038	0.904
White	23 (57.5)	17 (58.6)	6 (54.5)		
Black	1 (2.5)	1 (3.4)	0 (0.0)		
Asian	8 (20.0)	6 (20.7)	2 (18.2)		
Other	2 (5.0)	1 (3.4)	1 (9.1)		
Handedness, *n* (%)					
Right	37 (92.5)	27 (93.1)	10 (90.9)	0.055	1.000
Left	3 (7.5)	2 (6.9)	1 (9.1)		
Education, *n* (%)					
In college	31 (77.5)	22 (75.9)	9 (81.8)	0.162	1.000
In graduate school	9 (22.5)	7 (24.1)	2 (18.2)		
Marital Status, *n* (%)					
Single	36 (90.0)	26 (89.7)	10 (90.9)	0.412	0.814
Married	4 (10.0)	3 (10.3)	1 (9.1)		
Employment Status, *n* (%)					
Yes	30 (75.0)	23 (79.3)	7 (63.6)	1.045	0.418
No	10 (25.0)	6 (20.7)	4 (36.4)		
Clinical					
RPQ Total, Mean (SD)	12.58 (12.42)	16.76 (12.14)	1.55 (2.07)	6.505	<0.001
NSI Total, Mean (SD)	15.43 (14.06)	19.86 (13.91)	3.73 (4.65)	5.489	<0.001
Somatic/Sensory, Mean (SD)	5.58 (5.81)	7.34 (5.83)	0.91 (1.81)	5.304	<0.001
Cognitive, Mean (SD)	2.75 (2.88)	3.55 (2.89)	0.64 (1.50)	4.156	<0.001
Affective, Mean (SD)	7.10 (6.42)	8.97 (6.56)	2.18 (1.94)	5.023	<0.001
Injury Characteristics					
Number of Injuries, Mean (SD)		2.55 (1.33)	N/A		
Single Injury		9 (31.0)	N/A		
Multiple Injuries		20 (69.0)	N/A		
Time since the last Injury (years), Mean (SD)		4.48 (5.00)	N/A		
Mechanism of Injury, *n* (%)					
Sports-related		14 (48.3)	N/A		
Head hit		6 (20.7)	N/A		
High-level Falls		5 (17.2)	N/A		
Military-related		3 (10.3)	N/A		
Car accident		1 (3.4)	N/A		

mTBI, mild traumatic brain injury; BMI, body mass index; RPQ, Rivermead Post-Concussion Symptoms Questionnaire; NSI, Neurobehavioral Symptom Inventory.

**Table 2 biomedicines-10-00036-t002:** Dysregulated ExomiRNAs following Chronic mTBI.

Probe Name	Target Sequence	Log2FC	Adjusted *p*-Value
Upregulated			
hsa-miR-520e	AAAGUGCUUCCUUUUUGAGGG	0.98	0.03
hsa-miR-499b-3p	AACAUCACUGCAAGUCUUAACA	0.83	0.01
hsa-miR-520b	AAAGUGCUUCCUUUUAGAGGG	0.43	0.04
hsa-miR-4488	AGGGGGCGGGCUCCGGCG	0.42	0.03
Downregulated			
hsa-miR-625-5p	AGGGGGAAAGUUCUAUAGUCC	−0.96	0.08
hsa-miR-421	AUCAACAGACAUUAAUUGGGCGC	−1.39	0.08
hsa-miR-664a-3p	UAUUCAUUUAUCCCCAGCCUACA	−1.43	0.08
hsa-miR-28-3p	CACUAGAUUGUGAGCUCCUGGA	−1.49	0.04
hsa-miR-125a-5p	UCCCUGAGACCCUUUAACCUGUGA	−2.10	0.04
hsa-miR-222-3p	AGCUACAUCUGGCUACUGGGU	−2.14	0.09
hsa-miR-140-5p	CAGUGGUUUUACCCUAUGGUAG	−2.17	0.07
hsa-miR-98-5p	UGAGGUAGUAAGUUGUAUUGUU	−2.32	0.09
hsa-miR-148a-3p	UCAGUGCACUACAGAACUUUGU	−2.63	0.06
hsa-miR-423-5p	UGAGGGGCAGAGAGCGAGACUUU	−2.65	0.09
hsa-miR-107	AGCAGCAUUGUACAGGGCUAUCA	−2.75	0.07
hsa-miR-181a-5p	AACAUUCAACGCUGUCGGUGAGU	−2.81	0.09
hsa-miR-374a-5p	UUAUAAUACAACCUGAUAAGUG	−2.86	0.09
hsa-miR-340-5p	UUAUAAAGCAAUGAGACUGAUU	−2.87	0.07
hsa-miR-29b-3p	UAGCACCAUUUGAAAUCAGUGUU	−2.95	0.05
hsa-miR-191-5p	CAACGGAAUCCCAAAAGCAGCUG	−3.03	0.08
hsa-miR-199a-3p	ACAGUAGUCUGCACAUUGGUUA	−3.13	0.05
hsa-miR-126-3p	UCGUACCGUGAGUAAUAAUGCG	−3.13	0.09
hsa-miR-23a-3p	AUCACAUUGCCAGGGAUUUCC	−3.36	0.04
hsa-miR-142-3p	UGUAGUGUUUCCUACUUUAUGGA	−3.39	0.07
hsa-miR-223-3p	UGUCAGUUUGUCAAAUACCCCA	−3.62	0.04

ExomiRNA, exosomal microRNAs.

**Table 3 biomedicines-10-00036-t003:** Top IPA Biological Functions and Disease Pathway.

Diseases and Disorders	*p*-Value Range	Number of Molecules
Neurological disease	4.58 × 10^−2^–4.85 × 10^−14^	14
Organismal injury and abnormality	4.95 × 10^−2^–4.85 × 10^−14^	23
Psychological disease	4.58 × 10^−2^–4.85 × 10^−14^	13
Cancer	4.95 × 10^−2^–7.96 × 10^−13^	21
Reproductive system disease	4.85 × 10^−2^–1.49 × 10^−12^	20
**Molecular and Cellular Functions**	***p*-Value Range**	**Number of Molecules**
Cell cycle	4.02 × 10^−2^–2.46 × 10^−6^	4
Cellular movement	4.88 × 10^−2^–2.46 × 10^−6^	12
Cellular development	4.47 × 10^−2^–5.25 × 10^−6^	12
Cellular growth and proliferation	4.47 × 10^−2^–5.25 × 10^−6^	12
Cell death and survival	3.99 × 10^−2^–8.71 × 10^−5^	11
**Physiological System Development and Function**	***p*-Value Range**	**Number of Molecules**
Organismal development	3.74 × 10^−2^–6.45 × 10^−9^	11
Organismal functions	8.69 × 10^−4^–7.19 × 10^−4^	2
Tissue morphology	8.90 × 10^−5^–1.72 × 10^−3^	3
Hematological system development and functions	4.47 × 10^−2^–1.98 × 10^−3^	6
Immune cell trafficking	2.74 × 10^−2^–1.98 × 10^−3^	2

## Data Availability

Data that support the findings of this study are available upon reasonable request from any qualified investigator to the corresponding author.

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
