# Peer review of "Exosomal microRNA Differential Expression in Plasma of Young Adults with Chronic Mild Traumatic Brain Injury and Healthy Control"

_biomedicines, 2021, doi:10.3390/biomedicines10010036_

Round 1
Reviewer 1 Report
The authors have found differences in exosomal mRNA in subjects with mTBI and continuing symptoms compared to healthy controls. Pathway analysis suggests that these mRNA may be involved in neuro-inflammatory processes.
The writing is clear and concise and the study is well-executed. Although the sample size is small, the results would be significant if replicated in a larger sample.
A few comments:
1) The authors should explain how the subjects were recruited. Was it a general call to college students to participate in research or a specific recruitment strategy used.
2) Were all mTBI subjects symptomatic? Were asymptomatic chronic mTBI subjects excluded. What percentage of subjects had a loss of consciousness with mTBI (usually a marker for a more severe mTBI)?
3) Please specify both build and version of SPSS (current is 28)
4) Methods should explain how RPQ and NSI were administered and reference tests and spell out names (names and references not given in text). Again were all mTBI subjects symptomatic?
5) As a minor stylistic point, it is now customary to capitalize all racial and ethnic groups including White so as to not imply bias.
Author Response
We would like to thank both reviewers for their thoughtful comments, which had helped us a lot in refining our paper. We posted our point‐to‐point responses to each of the two reviewers, which all correspond to the revised manuscript. We indicated where we made changes by using the page and line numbers, in addition to using track changes in the main manuscript.
Reviewer 1: The authors have found differences in exosomal mRNA in subjects with mTBI and
continuing symptoms compared to healthy controls. Pathway analysis suggests that these mRNA may be
involved in neuro-inflammatory processes.
The writing is clear and concise and the study is well-executed. Although the sample size is small, the
results would be significant if replicated in a larger sample.
We thank the reviewer for the thoughtful comments.
Reviewer 1-1: A few comments: The authors should explain how the subjects were recruited. Was it a
general call to college students to participate in research or a specific recruitment strategy used.
We have now provided more information on how the participants were recruited in the
methods section to improve the clarity. The following sentence has been added to the text:
(In lines #61-63) “Voluntary participants of this study were recruited via flyers and email
advertisements to the members of the UNLV campus, including the Military and Veteran
Services Center, and the Las Vegas community.”
Reviewer 1-2: Were all mTBI subjects symptomatic? Were asymptomatic chronic mTBI subjects excluded.
What percentage of subjects had a loss of consciousness with mTBI (usually a marker for a more severe
mTBI)?
In our analysis, we included participants with and without symptoms. Post-injury symptoms
were analyzed based on the total sum cores of each questionnaire for NSI and RPQ. Higher
scores indicate more severe symptoms. We now added the detailed information of symptoms
assessment to the method section for clarity.
Detail of these post-mTBI symptoms reported was added into the method section as follows:
(In lines #67-78) “The Neurobehavioral Symptom Inventory (NSI) and the Rivermead Post-
Concussion Symptoms Questionnaire (RPQ) were used to assess the participant’s symptom
experiences following mTBI. The NSI consists of 22 symptoms on a Likert scale 0 to 4 (0 =
none, 1= mild, 2 = moderate, 3 = severe, and 4 = very severe) with a total sum score range of 0
to 88, and can be divided into three subscales affective, cognitive, and somatic/somatosensory
[18]. The NSI has good reliability and validity in measured post-concussive symptoms in TBI
patients [19]. The RPQ is 16-items questionnaires with a Likert scale of 0 to 4 (0 = none, 1=
mild, 2 = moderate, 3 = severe, and 4 = very severe) with total sum score range of 0 to 64. The
RPQ has an excellent internal consistency in TBI patients at all levels of severity—mild,
moderate, and severe TBI [20]. Higher total scores of NSI and RPQ indicate more severe
symptoms.”
We also added a sentence in the results regarding the participants with either loss of
consciousness or posttraumatic amnesia, as follows:
(in lines #149-152) “Only 1 participant in the mTBI group reported that they experienced both
a brief loss of consciousness and posttraumatic amnesia at the injury (3.4%). Two additional
participants with a history of mTBI reported having posttraumatic amnesia (6.9%).”
Reviewer 1-3: Please specify both build and version of SPSS (current is 28).
We have now provided the version of SPSS (in line #132).
Reviewer 1-4: Methods should explain how RPQ and NSI were administered and reference tests and spell
out names (names and references not given in text). Again were all mTBI subjects symptomatic?
We thank the reviewer for pointing out this important issue. As addressed above for the
Reviewer 1-2 comment, we have provided information on NSI and RPQ in the methods section
(In lines #67-78).
In our analysis, we included participants with and without symptoms. Post-injury symptom
levels were measured and analyzed based on the total sum cores of each questionnaire for NSI
and RPQ. Higher scores indicate more severe symptoms.
Reviewer 1-5: As a minor stylistic point, it is now customary to capitalize all racial and ethnic groups
including White so as to not imply bias.
We have now capitalized “White” in our revised manuscript (in lines #140 and 275)
Reviewer 2 Report
In this work, Vorn et al. evaluate the exosomal microRNAs (exomiRNAs) in plasma from young adults with or without a chronic mTBI. The authors found some microRNAs in plasma related to neuronal inflammation. The authors claim that TBI is associated with dysregulated exomiRNAs in the plasma-based on their results.
The article has a potential interest. Nevertheless, several major vital points must be addressed before final acceptance in Biomedicines.
- In the Introduction you definite the chronic mild traumatic brain injury: "Repetitive mTBIs, which usually contribute to chronic mTBI with unresolved and persistent symptoms, are common in athletes and military personnel, and these individuals have a higher risk of chronic neurologic impairment [4,5]”. But in Discussion part, you wrote that:” We were also unable to differentiate plasma exomiRNAs expression in single or multiple injuries in our small cohort.” Also, in the method part in table 1, you defined "Mechanism of Injury”. I do not think that you can use the termini “chronic mild traumatic brain injuries" because all experimental participants were not with repetitive mTBI in this study. You must indicate how many participants had repeated brain injuries.
- This study has some limitation factors:
- the small number of participants in the control group;
- there are no members of the black race in the control group
Therefore, the discussion section should be supplemented, and the limiting factors of this study clearly indicated.
- The results should be supplemented by analysis of a microRNA comparison between women and men.
- The supplementary video file is not included in the supplemental materials, and the video is not mentioned anywhere in the main manuscript.
Author Response
We would like to thank both reviewers for their thoughtful comments, which had helped us a lot in refining our paper. We posted our point‐to‐point responses to each of the two reviewers, which all correspond to the revised manuscript. We indicated where we made changes by using the page and line numbers, in addition to using track changes in the main manuscript.
Reviewer 2: In this work, Vorn et al. evaluate the exosomal microRNAs (exomiRNAs) in plasma from
young adults with or without a chronic mTBI. The authors found some microRNAs in plasma related to
neuronal inflammation. The authors claim that TBI is associated with dysregulated exomiRNAs in the
plasma-based on their results.
The article has a potential interest. Nevertheless, several major vital points must be addressed before final
acceptance in Biomedicines.
We thank the reviewer for the thoughtful comment.
Reviewer 2-1: In the Introduction you definite the chronic mild traumatic brain injury: "Repetitive
mTBIs, which usually contribute to chronic mTBI with unresolved and persistent symptoms, are common
in athletes and military personnel, and these individuals have a higher risk of chronic neurologic
impairment [4,5]”. But in Discussion part, you wrote that:” We were also unable to differentiate plasma
exomiRNAs expression in single or multiple injuries in our small cohort.” Also, in the method part in table
1, you defined "Mechanism of Injury”. I do not think that you can use the termini “chronic mild traumatic
brain injuries" because all experimental participants were not with repetitive mTBI in this study. You
must indicate how many participants had repeated brain injuries.
We thank the reviewer for the comments.
In this cohort, repetitive mTBI was included in the analysis. Repetitive mTBI has been
defined based on a number of injuries. We have reported that 65.5% of the chronic mTBI group
has multiple injuries, which can refer to repetitive mTBIs. The repetitive mTBI is presented in
Table 1 as multiple injuries.
For a clarity, we revised the sentence in the discussion as follows:
(In lines #279-280) “We were also unable to differentiate plasma exomiRNAs expression in
single or multiple injuries due to our small sample size.”Reviewer 2-2: This study has some limitation factors:
• the small number of participants in the control group;
• there are no members of the black race in the control group
Therefore, the discussion section should be supplemented, and the limiting factors of this study clearly
indicated.
The following statement was added in the limitation of this study:
(in lines #274-275) “The majority of our sample population consisted of White which limits
the genetic diversity and generalizability to all chronic mTBI populations.”
Reviewer 2-3: The results should be supplemented by analysis of a microRNA comparison between
women and men.
We believe the reviewer is bringing a very important point about sex differences. We believe
that sex differences exist in the mTBI population. Our recent articles show that men and women
have significant differences in biological and behavioral outcomes following mTBI in military
personnel (Sass et al., 2021). Investigating sex differences in exomiRNAs expression is needed to
develop a potential target treatment guideline for TBI. However, this manuscript is not focusing
on sex differences. In addition, we have a small sample size in this cohort, especially the control
group (Males = 4, Females = 7). A sample size that is too small will increase the false-positive
results which affect the implication of the study (Faber & Fonseca, 2014). Also, many
confounding factors (i.e., race, mechanism of injury, number of injuries) may mask the
significance of chronic mTBI. This is an important point that we could add to our future cohort
with enough sample size. Therefore, the following sentence was added into a limitation section,
as addressed above:
(in lines #274-275) “The majority of our sample population consisted of White which limits the genetic diversity and generalizability to all chronic mTBI populations.”
Reviewer 2-4: The supplementary video file is not included in the supplemental materials, and the video is
not mentioned anywhere in the main manuscript.
We thank the reviewer for pointing this out. We do not have a supplementary video to
include in this study. This statement was removed now from our manuscript.
References for responses
Sass, D.; Guedes, V.A.; Smith, E.G.; Vorn, R.; Devoto, C.; Edwards, K.A.; Mithani, S.; Hentig, J.; Lai, C.;
Wagner, C.; et al. Sex Differences in Behavioral Symptoms and the Levels of Circulating GFAP,
Tau, and NfL in Patients With Traumatic Brain Injury. Front. Pharmacol. 2021, 12, 3367,
doi:10.3389/fphar.2021.746491.
Faber, J.; Fonseca, L.M. How sample size influences research outcomes. Dental Press J. Orthod. 2014,
doi:10.1590/2176-9451.19.4.027-029.ebo
Round 2
Reviewer 2 Report
Thank you for your answer!